# The citrus tristeza virus p33 protein functions as a viroporin

**Vicken Aknadibossian**[1¤], **Clare Stokes**[2], **Roger L. Papke**[2], **Hao Wei Teh**[1], **Ying Wang**[1], **Svetlana Y. Folimonova**[1]*

**1** Department of Plant Pathology, Institute of Food and Agricultural Sciences, University of Florida, Gainesville, Florida, United States of America, **2** Department of Pharmacology and Therapeutics, College of Medicine, University of Florida, Gainesville, Florida, United States of America

¤ Current address: Department of Biology, Leuven Plant Institute, KU Leuven, Leuven, Belgium
* svetlana@ufl.edu

## Abstract

Viroporins are viral proteins that form ion channels in the membranes of the host and, thus, alter the host ion homeostasis to create more favorable environments for the virus. Since the discovery of the ion channel activity of the M2 protein encoded by influenza virus A (species: *Alphainfluenzavirus influenzae*), many additional viral proteins have also been characterized as viroporins. However, most viroporins known thus far belong to animal viruses, while the discovery of plant virus viroporins has significantly lagged. In this work, we present evidence that the p33 protein, a membrane-associated protein of citrus tristeza virus (CTV; species: *Closterovirus tristezae*), possesses the characteristics of a viroporin. We first show the substantial structural similarities between the transmembrane and cytoplasmic domains of known Class I viroporins and those of the p33 protein. Using two-voltage electrode clamp assays in *Xenopus* oocytes, we further demonstrate the ion channel properties of p33 such as the ability to induce strong inward currents of potassium and sodium when activated at lowered membrane potentials. Finally, using confocal and electron microscopy, we show that, similarly to other Class I viroporins, the p33 protein triggers extensive membrane remodeling and discuss additional characteristics of p33 and the functions of this protein in the CTV infection, which resemble those found with viroporins of other viruses. This study represents the third report of a viroporin encoded by a plant virus and the first validation of the ability of a plant virus viroporin to induce currents across eukaryotic membranes using electrophysiological assays. The findings of this work open new avenues in research focusing on the understanding the role of viroporins in plant virus infections.

provided the original author and source are credited.

**Data availability statement:** The manuscript contains all data required to replicate the results of this study.

**Funding:** This work was supported by the National Science Foundation (Grant Numbers MCB-1615723 and MCB-2316587 to S. Y. F.) and the United States Department of Agriculture (USDA) National Institute of Food and Agriculture (NIFA), Hatch Project FLA-PLP-006024 (to S. Y. F.). The funders had no role in study design, data collection and analysis, decision to publish, or preparation of the manuscript. Any opinions, findings, and conclusions or recommendations expressed in this material are those of the author(s) and do not necessarily reflect the views of the corresponding funding agencies.

**Competing interests:** The authors have declared that no competing interests exist.

## Author summary

Viroporins are known to influence viral pathogenicity, and, thus, a significant amount of research on clinically important human viruses has been focused on targeting these proteins to generate antiviral therapeutics. On the other hand, viroporins encoded by plant viruses are largely unexplored. This work represents a milestone in virology research. Here, we provide evidence that the p33 protein, a pathogenicity determinant of citrus tristeza virus (CTV), which represents one of the most economically important plant viruses, is a novel viroporin and induces currents across eukaryotic membranes in electrophysiological assays. Furthermore, we reveal new forms of the plant cell membrane remodeling induced by p33, which supports the notion that this CTV protein functions in a manner typical for known viroporins shown to induce similar forms of membrane alterations in animal cells. Finally, we discuss the functions of p33 in the CTV infection, which resemble those found for viroporins of other viruses. The findings of this work reveal novel aspects in virus-host interactions and open new avenues in the development of means to mitigate viral infections.

## Introduction

The first report of a virus-encoded ion channel protein, the M2 protein of influenza virus A (IAV; species: *Alphainfluenzavirus influenzae*), in 1992 [1] marked the beginning of research on an exciting and previously unknown class of viral protein: viroporins. Over the years, many clinically important animal viruses were shown to encode such proteins. Viroporins or viral ion channel forming proteins are transmembrane proteins that oligomerize in the cellular membranes and form ion channels, which allow selective movement of certain ions such as potassium, sodium, or calcium [2,3]. Viroporins play diverse roles in the virus infection cycle and have been found to regulate many viral processes and host responses, influencing virus pathogenicity [3,4]. Thus far, viroporins have been reported to facilitate viral entry and uncoating (e.g., the IAV M2 protein), viral replication and assembly (e.g., the rotavirus NSP4 protein), and viral release (e.g., human immunodeficiency virus type 1 (HIV-1) (species:*Lentivirus humimdef1*) Vpu protein). Viroporins also regulate key host antiviral responses such as autophagy (e.g., the IAV M2 protein), apoptosis (e.g., the severe acute respiratory syndrome-related coronavirus (SAR-CoV; species: *Betacoronavirus pandemicum*) envelope (E) protein), and immune responses (e.g., the HIV-1 Vpu protein) (reviewed in [4]). The deletion of a viroporin-encoding gene from a viral genome is often detrimental to the virus [2]. For this reason, there has been considerable work attempting to identify channel blockers that would nullify the effect of the viroporins encoded by clinically important viruses and work as viral therapeutics [5]. It should be noted, however, that many viroporins are multifunctional proteins, and many of their functions in the viral infection cycle are independent of the ion channel activity of their transmembrane domains [6,7]. Viroporins are also referred to as viral

channel forming proteins (VCPs) in the literature. It can be argued that VCP may be a more fitting name for these proteins as viroporins do not share the beta barrel motif characteristic of "porins", which they are named after [8,9]. However, the term viroporin has prevailed over the years and is the commonly used name in literature.

Since the discovery of the M2 viroporin, puzzling functions of viroporins have been increasingly investigated, and viroporins have been discovered for more than 30 different viruses from multiple different families (reviewed in [4]). Interestingly, the vast majority of the reported viroporins belong to animal-infecting viruses, with the exception of the eukaryotic algae-infecting paramecium bursaria chlorella virus (PBCV-1; species: *Chlorovirus vanettense*) [10,11]. This raised the question of whether this peculiar class of proteins, which are fairly common among animal viruses and are even reported in algae viruses, have been overlooked in the diverse and widely studied plethora of plant viruses. Remarkably, however, two recent reports presented data suggesting that the potyvirus 6K1 protein [12] and the rhabdovirus P9 protein [13] function as viroporins. While ample evidence was provided to support such function of these proteins, neither study demonstrated the ion channel activity in electrophysiological assays that have been used as classic confirmatory approaches required to establish a protein as a viroporin [14]. Nonetheless, accumulating evidence suggests that viroporins are ubiquitous among viruses infecting all forms of life and play significant roles in enhancing viral fitness. Further advancements in this field hold great potential to deepen our understanding of host-virus interactions and develop novel means to mitigate the threat of agriculturally or clinically important viruses.

Citrus tristeza virus (CTV; species: *Closterovirus tristezae*; genus: *Closterovirus*; family: *Closteroviridae*) is the most economically important viral pathogen of citrus, which has caused significant yield losses and tree death in many citrus-growing regions around the world [15]. The 19.3-kilobase CTV genome is the largest among non-segmented RNA genomes of plant viruses and harbors 12 open reading frames (ORFs) [16]. The 5' proximal replication block consists of ORF1a and ORF1b. ORF1a encodes a polyprotein containing two leader proteases plus methyltransferase- and a helicase-like domains. An occasional +1 ribosomal frameshift upon translation of ORF1a results in the continuation of the protein synthesis through ORF1b, leading to the expression of the polymerase domain [16]. The genome of CTV also contains a five-gene block, which includes the ORFs for p6, p65, p61, p27 (the minor coat protein), and p25 (the major coat protein) and is conserved among the members of the family *Closteroviridae*. Apart from p6, which is needed for virus movement, the other four proteins are required for efficient virion assembly [17,18]. The p20 and p23 proteins whose corresponding ORFs are positioned in the 3'-terminal portion of the virus genome act as the CTV viral suppressors of RNA silencing (VSRs), along with the major coat protein [19]. Finally, there are three additional nonconserved ORFs encoding the p13, p18, and p33 proteins that are unique to CTV with no significant sequence homology to any known proteins [20].

The p33 protein is a mystifying protein of CTV. Multiple roles have been assigned to p33, and the protein has been implicated in diverse viral processes. The protein possesses a C-terminal transmembrane domain (TMD) and appears to be an unconventional virus movement protein [21,22]. It was shown to localize to the plasmodesmata channels connecting plant cells, form extracellular tubules in protoplasts, and utilize the cellular secretory pathway for its intracellular trafficking. Furthermore, p33 colocalizes with p6, a small hydrophobic movement protein of CTV [22]. Although a mutant virus with the deletion of the p33 gene can establish systemic infection in a number of citrus varieties as well as in an herbaceous host, *Nicotiana benthamiana* [20], lack of p33 results in a significant delay of the virus systemic infection, supporting its role in the *in planta* systemic spread of CTV [23]. Moreover, the p33 protein is absolutely needed for infection of a few other virus hosts, and the membrane association of p33 is crucial for its ability to mediate such infection [21,24]. In addition, p33 functions as a determinant of CTV superinfection exclusion, a phenomenon in which a primary virus infection prevents a secondary infection of the host with the same or closely related virus [25,26]. The p33 protein interacts with a viral long noncoding RNA, LMT1, and both serve as plant immunity modulators [23,27,28].

Remarkably, certain characteristics of p33 mentioned above such as its membrane association, the ability to induce the formation of membranous structures, and its effect on the virus pathogenicity resemble those of known viroporins. Such parallels inspired us to examine the possibility that p33 functions as an ion channel protein. In this work, we unraveled

extensive structural similarities between the transmembrane and cytoplasmic domains of known Class I viroporins and those of the CTV p33 protein. We further demonstrated that p33 is indeed a functional viroporin in two-voltage electrode clamp assays in *Xenopus* oocytes with strong inward currents of potassium and sodium when activated at lowered membrane potentials. Similarly to other Class I viroporins, p33 was also shown to induce extensive membrane remodeling independent of virus-infection. Our findings represent the first validation of a plant virus viroporin activity in an electrophysiological assay and open new avenues of research on the role of viroporins in plant virus infections.

## Results

### p33 shares structural characteristics with Class I viroporins

The identification of novel viroporins is challenging as there is a lack of sequence homology between viroporins, and viroporins of different viruses could play distinct roles in the infection cycles of the respective viruses [14]. However, there are some structural and functional characteristics shared by viroporins, which aid in making predictions on which viral proteins could act as viroporins [2,4,14,29]. Accordingly, to explore whether the p33 protein could be a viroporin, we first compared its structural organization to that of known viroporins. The classification system for viroporins takes into account the number of TMDs they possess and classify the proteins into three classes: Class I (one TMD), Class II (two TMDs), and Class III viroporins (three TMDs) [2,4]. It has previously been demonstrated that p33 contains a single C-terminal TMD [21]. Therefore, the position of the TMD within the protein sequence of p33 and the amino acids adjacent to the TMD were compared to those of known Class I viroporins such as the M2 protein of IAV, HIV-1 Vpu, and the SARS-CoV E proteins, which has revealed several similarities (Fig 1A). The three viroporins and p33 all possess a relatively long cytoplasmic domain (CD) and a much shorter external domain (ED), which is only a few amino acids long in the case of the p33, Vpu, and E proteins. The TMD, CD, and ED domains of the viroporins mentioned above have been determined in previous studies [5,21,30,31]. Furthermore, similar to other viroporins shown to harbor basic amino acid residues (such as lysine and arginine) adjacent to their channel-forming TMD [4,14,30], the p33 protein also contains lysine and arginine residues within the portion of its cytoplasmic domain connected to its TMD (the respective residues in p33 and the other displayed viroporins are highlighted in red in Fig 1A).

   All known viroporins have been shown to self-oligomerize in the host membranes as either tetramers, pentamers, or hexamers to form ion channels [29]. Previous work in our lab demonstrated the self-interaction of p33 through co-immunoprecipitation (Co-IP), yeast-two-hybrid (Y2H), and bimolecular fluorescence complementation assays (BiFC), but the study did not explore possible oligomerization beyond dimers [32]. To investigate whether p33 forms higher oligomers, a western blotting analysis with an anti-p33 antibody was performed using crude extracts from the *N. benthamiana* leaves agroinfiltrated with either a p33-expressing ("p33") cassette or an empty vector ("EV") construct. The "p33" lane in the western blot revealed three distinct bands with the molecular weights corresponding to the monomeric (33 kDa), dimeric (66 kDa), and tetrameric (132 kDa) forms of p33, while they were absent in the samples from leaves harboring an empty vector as a negative control (Fig 1B). This finding suggested that p33 can form stable dimers and tetramers *in planta*. In an effort to obtain a clearer blot in which the bands are less diffused and permit better approximation of their corresponding molecular weights, the p33 protein ectopically expressed in the *N. benthamiana* leaves was immunoprecipitated from a leaf extract using magnetic protein A beads conjugated to the anti-p33 antibody, following the procedure described previously [21]. The precipitated protein was eluted in the SDS loading buffer supplemented with 8 M urea and analyzed by western blotting using the anti-p33 antibody, which revealed clearly distinct bands of the molecular weights corresponding to the monomeric (33 kDa), dimeric (66 kDa), and tetrameric (132 kDa) forms of p33 (S1 Fig), supporting the ability of the protein to oligomerize. The oligomeric forms of p33 remained under the denaturing conditions of the sodium dodecyl sulfate (SDS) buffer supplemented with 100 mM dithiothreitol (DTT) and 8 M urea. Interestingly, this has also been observed for ligand-gated ion channels such as the nicotinic α7 and 5-hydroxytryptamine 3A receptors, which

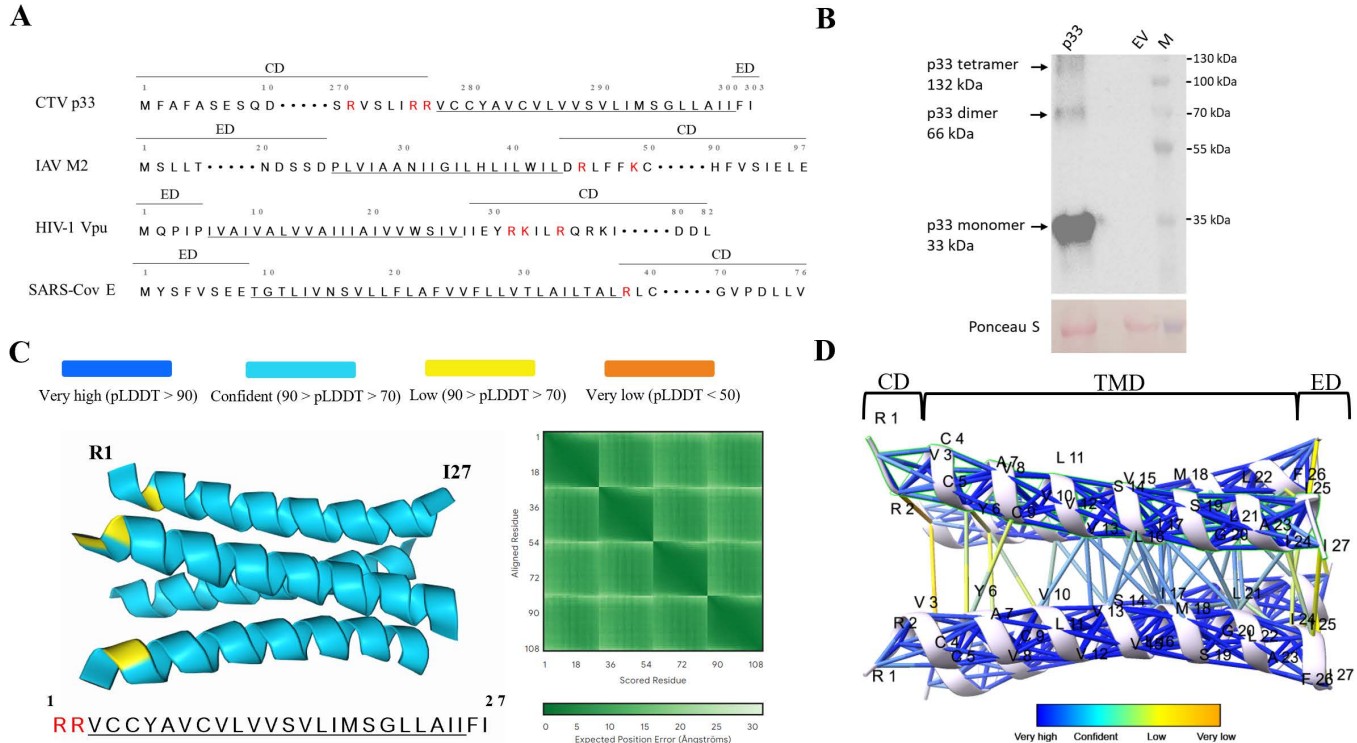

**Fig 1. Structural Class I viroporin characteristics displayed by p33.** A) The TMD of p33 compared with that of three known viroporins (IAV M2, HIV-1 Vpu, and SARS-CoV E) representing the underlined TMD, the short external domains (ED) and longer cytoplasmic domains (CD), with cytoplasmic TMD-adjacent basic residues colored in red. B) Western blot of the crude extracts of *N. benthamiana* leaves agroinfiltrated with either a p33-expressing (p33) or an empty vector (EV) construct carried out with an anti-p33 primary antibody. The p33 loaded lane displayed protein bands, which correspond to the expected sizes of p33 monomers, dimers, and tetramers. Ponceau S staining was used to display the amount of loaded protein in the p33 and EV lanes. M: Molecular marker. C) AlphaFold3 prediction of the tetrameric p33 TMD with the two amino acids on the preceding CD and succeeding ED domains. The blue color of the helices indicates a confident pIDDT score for the p33 TMD and the PAE heatmap displays the confidence of the position of the four p33 TMD monomers with respect to each other in the model. D) AlphaFold3 predicted aligned error for the residues at the binding interface between the p33 TMD monomers.

also show multiple order of homo-oligomerization in the presence of SDS, 8M urea, and 100 mM DTT, similar to p33 [33]. To further assess the p33 structure, we used AlphaFold 3, a leading protein structure prediction tool, which is characterized by unprecedented high accuracy in predicting biomolecular interactions, including protein-protein complexes [34]. To examine if the TMD of p33 shows any resemblance to the structures of known viroporin TMDs, a tetrameric input of the p33 TMD amino acid sequence with the addition of the two flanking external and cytoplasmic amino acids (RRVC-CYAVCVLVVSVLIMSGLLAIIFI; underlined sequence corresponds to TMD) was analyzed with AlphaFold 3 (Fig 1C). The predicted structure displayed confident predicted local distance difference test (pLDDT) and predicted aligned error (PAE, a measure of the expected error in the position of one residue with respect to another residue in a model) scores and placed the four p33 TMD chains in close proximity in a configuration that was highly reminiscent of the experimentally determined structures of known viroporin TMD channels [2,29] (Fig 1C). The examination of the PAE confidence of the binding interface between residues within each individual chain and between chains revealed that the structure prediction of each individual chain was highly confident (shown by the dark blue connections between the amino acids within each chain in Fig 1D). As for the prediction confidence of the positions of the chains relative to each other, the prediction was fairly confident for the residues in the TMD region and less confident for the external TMD-adjacent amino acids on both

ends (Fig 1D). Altogether, the above structural characteristics suggested that p33 forms homo-oligomeric tetramers in which the TMDs are predicted to line up in a viroporin channel-like configuration.

## Prediction of a possible p33 channel pore

Given the channel-like configuration of the p33 TMD predicted structure, we next attempted to predict the structure of a pore that could be formed by TMDs of the oligomerized p33 subunits. When examining the surface of the predicted p33 tetrameric TMD channel from the cytoplasmic and external ends, a narrow channel-like opening that stretches from end-to-end was observed, with constrictions at the Y6 and L21 residues (Fig 2A). To further explore the characteristics of the channel, the MOLEonline tool was applied to predict a channel-spanning pore [35]. The software identified a 46 Å-long pore (Fig 2B) lined with the residues R2, Y6, V13, I17, L21, and I24 (Fig 2C). The bottleneck constriction point of the predicted pore appears to be at L21, with a radius of 1 Å (Fig 2B). The radius of the pore signifies a sphere within

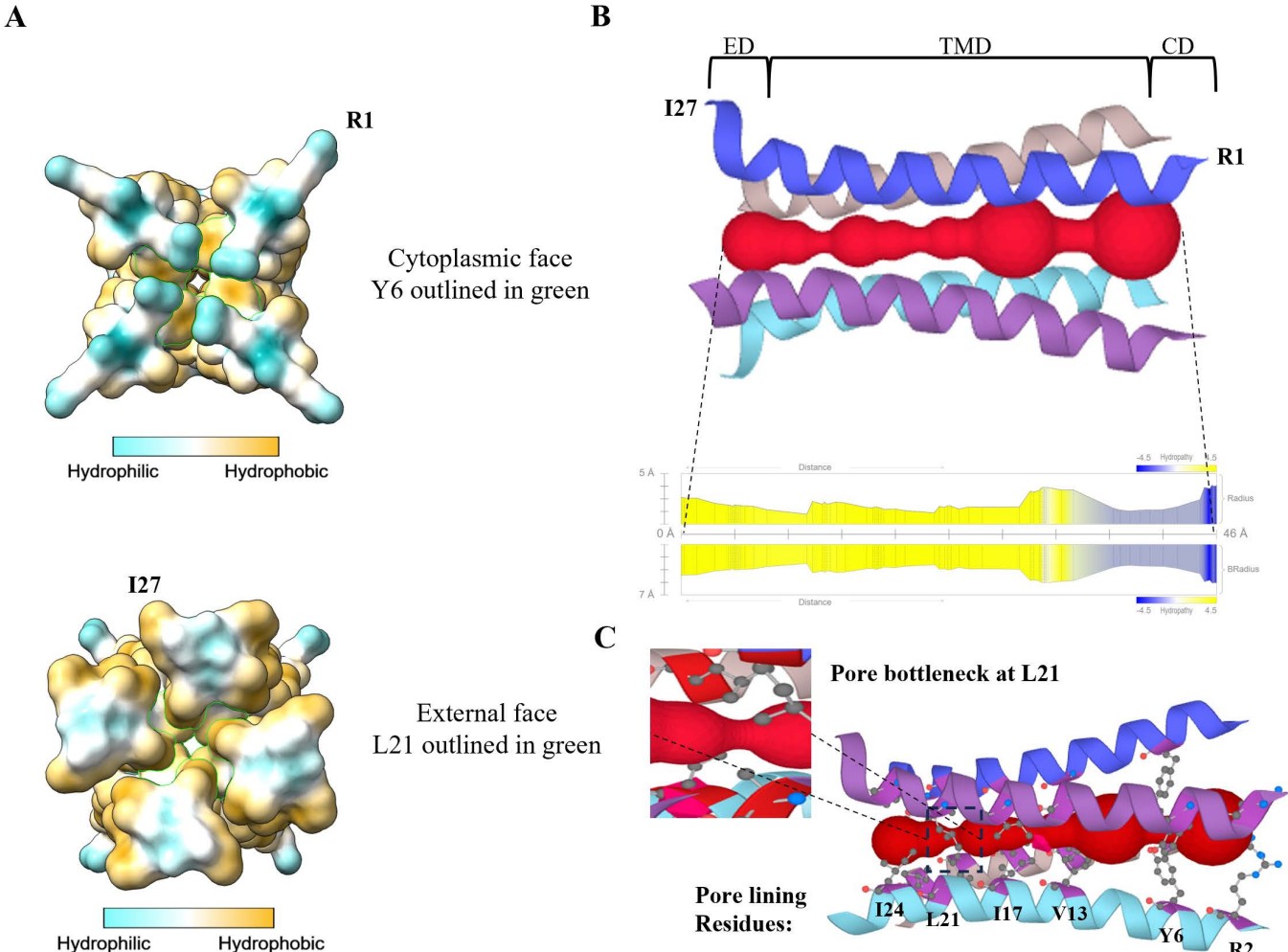

**Fig 2. Prediction of p33 TMD pore structure and surface hydrophobicity.** A) Surface view of the cytoplasmic and external faces of the predicted structure of p33 TMD depicting surface hydrophobicity. The pore lining residues Y6 and L21 are outlined in green. B) The predicted structure of a pore spanning the p33 TMD with a length of 46 Å and a bottleneck radius of 1 Å at L21. The graph illustrates the radius, Bradius, and the hydrophobicity of the pore along the residues of the p33 TMD. C) Depiction of the side chains of the pore lining residues of the p33 TMD predicted pore and a zoomed-in view of the L21 bottleneck.

the pore limited by the three closest atoms, which are the side chains of L21 for the narrowest constriction point (Fig 2C). An alternative measure, the Bradius of a pore, accounts for the local flexibility of the residues by calculating the Root-Mean-Square-Fluctuations (RMSF) of the amino acids using the B-factors of the structures and adds it to the calculated radius [36]. The RMSF is a measure of the displacement of a particular atom or a group of atoms relative to the reference structure, averaged over the number of atoms [37]. Using Bradius, the bottleneck of the pore at L21 increased to 3.5 Å (Fig 2B). Although the predicted pore is not yet experimentally validated, and channel proteins often have open and closed states with distinct protein conformations, it provides a theoretical model of a possible p33 channel pore structure.

The pore bottleneck radius of the SARS-CoV-2 viroporin E was modelled to be 2.3 Å in its closed state and 5.6 Å in its open state [38]. A comprehensive study documented the differences in nuclear magnetic resonance structures of the open- and closed-state IAV M2 and influenza B virus (species: *Betainfluenzavirus influenzae*) M2 (BM2) viroporin TMDs [39]. The viroporins of these related viruses modify their structures very distinctly between open and closed states. Considering this, it would be difficult to predict how another viroporin TMD such as that of p33 would open, following the comparison with other viroporins. In the case of BM2, the transmembrane helices open in a scissor motion, increasing the tilt angle by six degrees and increasing the pore diameter between 1.4-2.8 Å along the pore lining residues [39]. As for eukaryotic ion channels, the cryo–electron microscopy (cryo-EM) structures of the human α7 nicotinic acetylcholine receptors (nAChR) had the narrowest pore constriction site radius of ~1.4 Å in the resting state, which expanded to around ~1.9 Å in the presence of a high affinity agonist [40]. Cryo-EM of the human inward-rectifier potassium channel Kir2.1 in its closed state revealed constriction sites with minimum pore radii of 0.77 Å and 1.05 Å [41]. Thus, the 1 Å calculated radius of the constriction sites of the predicted TMD pore of p33 seems to be in line with the pore radius of other potassium channels and viroporins in their closed states and it would probably enlarge as the channel is activated.

## The cytoplasmic domain of p33 displays characteristic similarities with those of Class I viroporins

In light of many structural features of the p33 TMD shared with Class I viroporins' TMDs, the cytoplasmic domain of the p33 protein was also compared to that of known viroporins in this class. Firstly, Class I viroporins often possess an amphipathic α-helix [4,42,43]. The amphipathic helix is in the cytoplasmic region of the HIV-1 Vpu protein [44] and IAV M2 protein [45]. An amphipathic helix has hydrophobic residues on one face of the helix and polar residues on the other face. Apart from this characteristic, they are quite diverse in the function, length, and composition of hydrophobic and polar residues [46]. This complicates the accurate prediction of amphipathic helices. Using the prediction tool AmphipaSeeK [47], we identified only a single stretch of six amino acid residues in p33 as possibly amphipathic (Fig 3A). To further examine this prediction, the predicted region and its surrounding amino acids were plotted on a helical wheel projection using Heliquest [48]. The 18 amino acid-long putative cytoplasmic α-helix, which includes amino acids 109–126 (RFVIRVKAVPASMRGYYS), showed a clear separation of hydrophobic and polar residues into two faces (Fig 3B).

Furthermore, Class I viroporins such as the HIV-1 Vpu, IAV M2, and SARS-CoV E proteins are phosphorylated at the cytoplasmic domain [2]. To examine whether p33 is phosphorylated as well, the p33 protein was ectopically expressed in the *N. benthamiana* leaves via agroinfiltration of a p33-expression cassette. The protein was immunoprecipitated from the leaf lysates using an anti-p33 antibody conjugated to protein A magnetic beads and submitted to a proteomics & mass spectrometry analysis for phosphopeptide identification. The S8 residue was identified as a phosphorylated site with a confident Mascot ion score and clear b- and y-ion peaks supporting the site's phosphorylation (Fig 3C). During two independent runs, four more putative phosphorylation sites (S120, T239, T241, and Y265) were computationally identified in p33. However, low Mascot ion scores and high noise to signal ratios of the peaks corresponding to these sites prevent us from discerning if they are truly phosphorylated or false positives. As such, we do not rule out the presence of other phosphorylated residues in the p33 cytoplasmic domain, in addition to S8.

Altogether, p33 is highly reminiscent of the Class I viroporins. It is a self-oligomerizing transmembrane protein with a small external domain and a larger cytoplasmic domain, with positively charged residues preceding the TMD. The

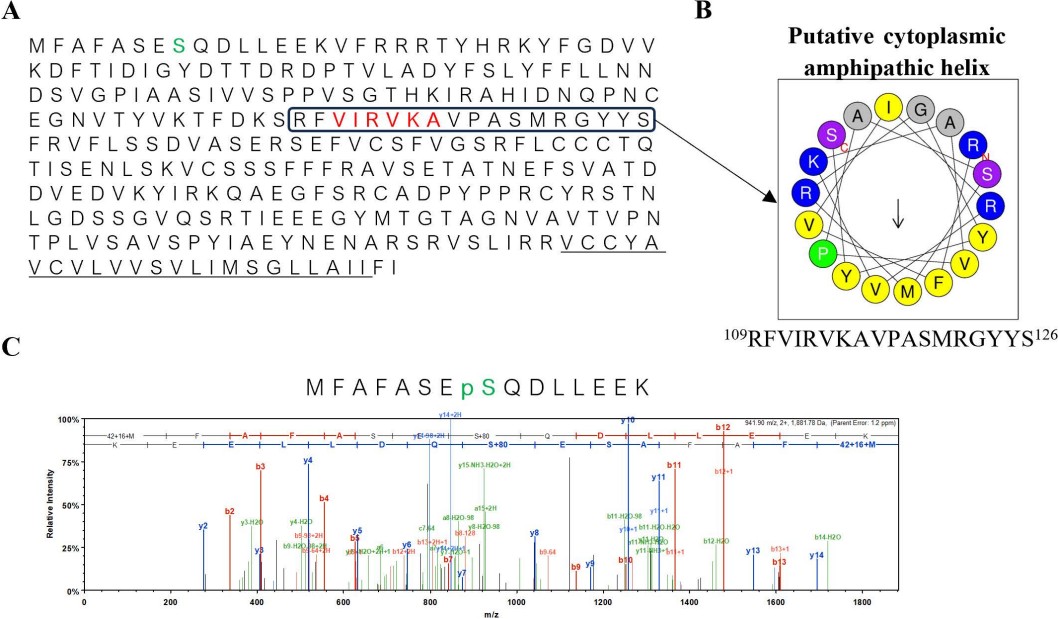

**A**

```
M F A F A S E S Q D L L E E K V F R R R T Y H R K Y F G D V V
K D F T I D I G Y D T T D R D P T V L A D Y F S L Y F F L L N N
D S V G P I A A S I V V S P P V S G T H K I R A H I D N Q P N C
E G N V T Y V K T F D K S R F V I R V K A V P A S M R G Y Y S
F R V F L S S D V A S E R S E F V C S F V G S R F L C C C T Q
T I S E N L S K V C S S S F F F R A V S E T A T N E F S V A T D
D V E D V K Y I R K Q A E G F S R C A D P Y P P R C Y R S T N
L G D S S G V Q S R T I E E E G Y M T G T A G N V A V T V P N
T P L V S A V S P Y I A E Y N E N A R S R V S L I R R V C C Y A
V C V L V V S V L I M S G L L A I I F I
```

**B**

**Putative cytoplasmic amphipathic helix**

$^{109}$RFVIRVKAVPASMRGYYS$^{126}$

**C**

M F A F A S E pS Q D L L E E K

**Fig 3. p33 displays cytoplasmic features of Class I viroporins.** A) The amino acid sequence of p33 portraying the underlined TMD, phosphorylated serine residue S8 in green, the Amphiseek (https://npsa-prabi.ibcp.fr/cgi-bin/npsa_automat.pl?page=/NPSA/npsa_amphipaseek.html; Sapay et al., 2006) predicted amphipathic domain in red, and the putative cytoplasmic amphipathic helix in a box. B) Helical wheel projections of a putative p33 cytoplasmic amphipathic helix portraying the orientation of the amino acid residues in an α-helix made using the Heliquest software (http://heliquest.ipmc.cnrs.fr; Gautier et al., 2008). Color coding for residues: yellow for hydrophobic, purple for Ser (S) and Thr (T), blue for Lys (K) and Arg (R), gray for small residues (Ala (A) and Gly (G)), and green for Pro (P). The cytoplasmic helix displayed distinct hydrophobic and hydrophilic sides of the helix, suggesting an amphipathic nature. C) MS/MS spectrum of a doubly charged peptide of m/z 941.90 with the b ions in red, the y ions in blue, and the ions with a neutral loss in green.

predicted TMD structure and the predicted pore within the structure line up well with those of experimentally verified Class I viroporins. Finally, the cytoplasmic tail of p33 is phosphorylated and possesses a putative amphipathic alpha helix. All these observations support the notion that p33 could be a possible Class I viroporin. We next followed with functional electrophysiological studies examining ion conductance, the test that is used as a classical approach to define a candidate protein as viroporin [43].

## Electrophysiological assays demonstrate p33 ion channel activity

The two-electrode voltage-clamp (TEVC) assay in *Xenopus laevis* oocytes is an established method to examine viroporin ion channel activity [14] and a suitable system to study plant ion channels [49]. To assess whether p33 is an ion channel protein, *Xenopus* oocytes were injected with either water as a control or approximately 27 ng of cRNA harboring the p33 coding sequence. Currents from water- and p33 cRNA-injected oocytes were elicited with 2-s voltage pulses at 10 mV increments ranging from +50 to -150 mV in 50 mM K+ Ringer's solution. The voltage was lowered to -150 mV as the membrane resting potential in plants is much more negative than that of oocytes and lower voltages might be necessary to elicit the current changes [49]. The current changes in plant potassium channels such as KAT1 start to appear around -100 mV and continue to intensify as the voltage is lowered down to around -200 mV [50]. The activation of inward currents of a plant phosphate transporter PHT1–6 only became significant between -130 and -160 mV [51]. However, membrane potentials of -160 mV and lower could destabilize the membrane and activate endogenous currents [49,52]. LaCl$_3$

has been used in some cases to block endogenous channels [50]. Nevertheless, we decided that it would be best to avoid this complication and not exceed -150 mV.

As the step changes in the membrane potential were lowered beyond -100 mV down to -150 mV, large inward currents were recorded in the p33-expressing oocytes (n = 14) as opposed to much smaller currents in water-injected oocytes (n = 7), suggesting that p33 forms functional ion channels (Fig 4A). Western blots using an anti-p33 antibody were performed with the total lysates of each of the 14 individually clamped oocytes confirming the p33 expression in the analyzed oocytes (Fig 4B). A second blot was performed comparing the bands observed in the lysates from the water-injected and the p33-injected oocytes, demonstrating the absence of the p33-specific band in the water-injected oocytes (Fig 4C). As can be seen in Fig 4A, the mean change ±SEM in the current induced in the p33-exprssing oocytes was significantly greater than that in the water-injected oocytes at -150 mV (p < 0.01). In conclusion, the difference in the current between the p33-expressing oocytes and the water-injected oocytes was substantial and significant, suggesting that p33 behaves like an inward rectifying potassium ion channel. The same assay was conducted in parallel in Ringer's solution (high in Na$^+$ and low in K$^+$), to determine if p33 also showed sodium conductance. Most viroporins that conduct potassium also conduct sodium, thus, it was not surprising that p33 also showed marked inward currents for Na+ (Fig 4D) [4].

**p33 induces extensive membrane remodeling that is reminiscent of Class I viroporins**

Multiple viroporins have been shown to participate in membrane remodeling [2]. The SARS-CoV-2 E viroporin induced the formation of small vesicles apparently budding from a large tubular lipid structure when incorporated into artificial phospholipid membranes that mimic intracellular biomembranes [53]. The IAV M2 viroporin was also demonstrated to induce the endosomal sorting complex required for transport (ESCRT)-independent and cholesterol-dependent budding in giant unilamellar vesicles through its cytoplasmic amphipathic helix [54,55]. The SARS-CoV viroporin ORF3a induced the accumulation of intracellular vesicles [56]. The HIV-1 Vpu viroporin induced the formation of uniflagellar enlarged vesicles (EVs) and also localized on the outer membrane of multivesicular bodies (MVBs) [57].

It has been shown previously that p33 induces the accumulation of vesicles, which colocalize with the endoplasmic reticulum (ER) and Golgi markers as well as with the endosomes, when transiently expressed in the *N. benthamiana* leaves [22]. It was postulated that p33 utilizes the cellular secretory pathways and endocytic-recycling pathway plus the actin network to reach the plasmodesmata [22]. These observations were carried out at a relatively early timepoint (four days post infiltrations (dpi)). In an effort to get a better look at how p33 remodels the cellular membrane, we attempted to monitor a green fluorescent protein (GFP)-tagged p33 (GFP:p33) ectopically expressed in the *N. benthamiana* epidermal cells at later stages of its expression. Previously, it was shown that the use of the p22 VSR of tomato chlorosis virus (species: *Crinivirus tomatichlorosis*) enhanced the expression of CTV in *N. benthamiana* [58]. We, therefore, engineered a binary vector expressing only p22 to use it as enhancer of the p33 expression in co-infiltration assays. Co-expression of p22 with GFP:p33 yielded improved protein expression and the GFP fluorescence, in comparison to that with other VSRs tested.

With the enhanced expression level, GFP:p33 induced the accumulation of vesicles similar to the ones previously described [22] (Fig 5A). In our study, GFP:p33 was also observed in MVBs (several single-membrane vesicles (SMVs) surrounded by a common membrane), and occasionally, in multiple MVBs in the same cell (Fig 5B). Interestingly, the GFP signal inside the MVBs appeared to be excluded from the smaller vesicles encapsulated in the MVB. The vesicles and MVBs were more concentrated near the cell wall. As fluorescence confocal microscopy could not provide higher resolution, transmission electron microscopy (TEM) was utilized to visualize transiently expressed untagged wild type p33 in the *N. benthamiana* epidermal cells.

Indeed, the TEM imaging revealed that the majority of the p33-induced vesicles were observed in close proximity to the cell wall (Fig 6A-E). Interestingly, they appeared to be extracellular as they were observed in the area between the plasma membrane and the cell wall (Fig 6A-E). Moreover, in the area where these vesicles were concentrated, the plasma

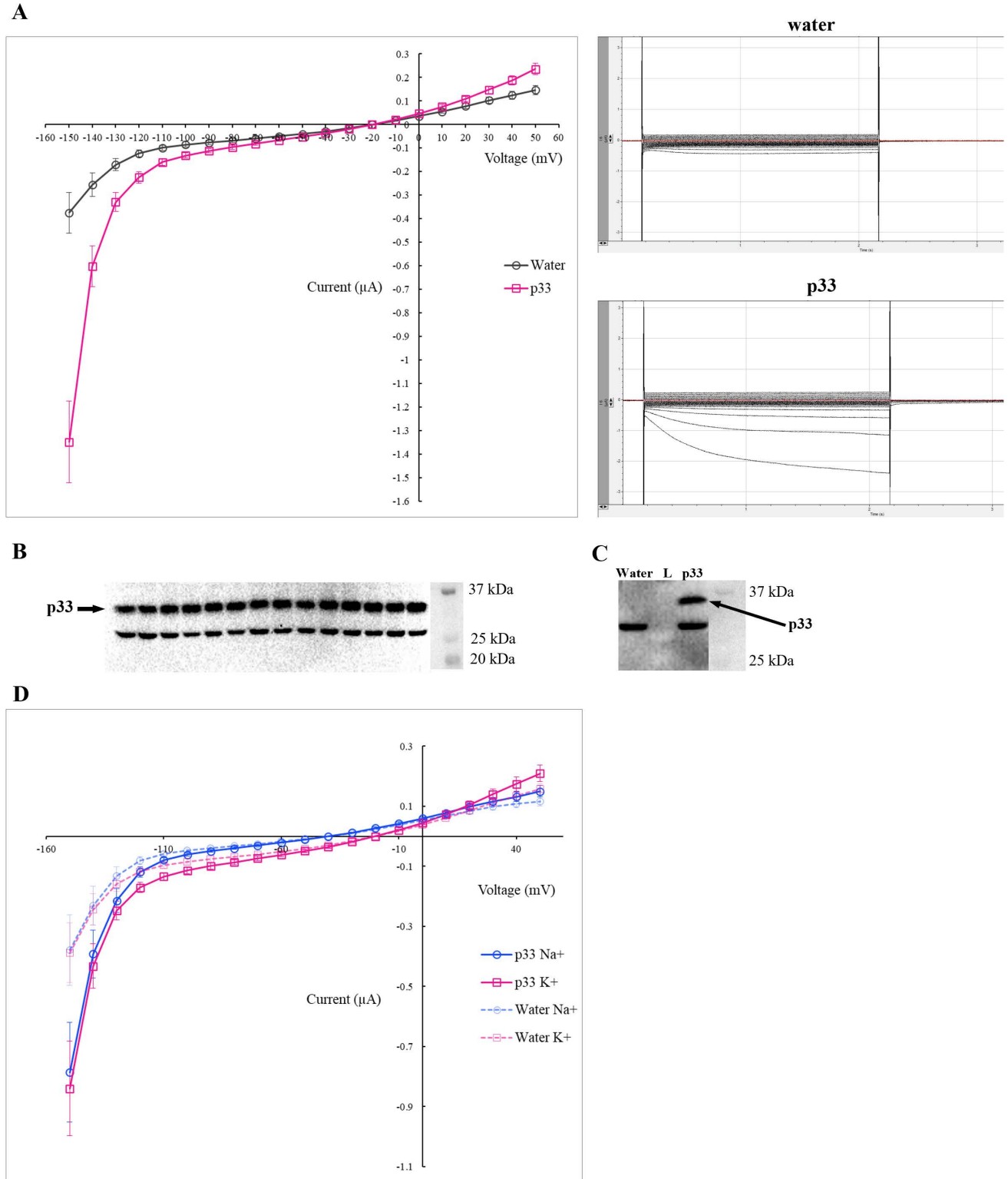

**Fig 4. p33 exhibits marked ion channel activity in *Xenopus* oocytes.** A) The TEVC current-voltage relationship (left) and example current traces (right) of water-injected (n = 7) and p33-injected (n = 14) oocytes at 10 mV incremental steps from + 50 to - 150 mV in 50 mM high K⁺ Ringer's solution 8 days after injection. Current values were averaged across all the western blot-positive oocytes tested, and the data represents the mean ± SEM. B)

Western blot performed with the anti-p33 primary antibody of the crude lysates of all 14 p33-injected oocytes tested. C) Western blot performed with the anti-p33 primary antibody of the lysates of a water-injected (water) and p33-injected (p33) oocytes. L: Ladder. D) The TEVC current-voltage relationship of currents in water-injected (n = 7) and p33-injected (n = 12) oocytes at 10 mV incremental steps from + 50 to - 150 mV in 50 mM high K⁺ Ringer's solution and normal Ringer's solution 6 days after injection. Current values were averaged across all the western blot-positive oocytes tested, and the data represents the mean ± SEM.

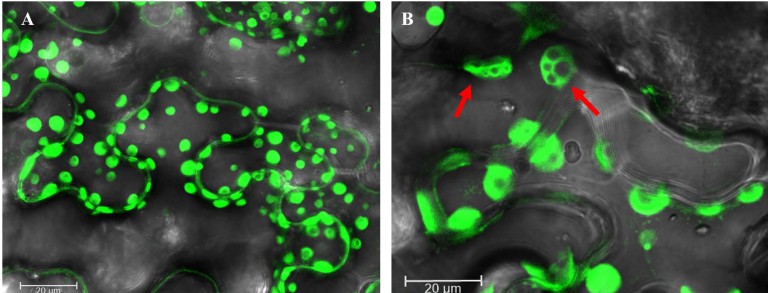

**Fig 5. The ectopic expression of GFP:p33 induced the accumulation of GFP:p33-containing vesicles and MVBs in *N. benthamiana* epidermal cells.** A) Epidermal cells were agroinfiltrated with a GFP:p33-expressing construct and visualized at four dpi using a confocal microscope. The cells exhibited numerous vesicles with the GFP fluorescence. B) Epidermal cells were agroinfiltrated with a GFP:p33-expressing construct and visualized at seven dpi using a confocal microscope. The cells show multiple GFP fluorescence-exhibiting vesicles and MVBs (red arrows).

membrane seemed to be pushed further away from the cell wall (Fig 6A-B; orange bars) when compared to other areas of the cell wall-plasma membrane interface (Fig 6A-B, blue bars). This observation signifies that p33 does not cause a general plasmolysis in the cells but rather leads to the localized pockets in which these vesicles accumulate. Most of these vesicles appear to be unilamellar or SMVs (Fig 6C), however, structures resembling double-membrane vesicles (DMVs) were also observed (Fig 6D). The size of the vesicles was highly variable, ranging between 50–1000 nm in diameter. Furthermore, MVBs were observed at the plasma membrane, appearing, in our interpretation, as been fused with the plasma membrane and releasing SMVs into the extracellular space (Fig 6E). Given the p33's association with the cellular secretory machinery [22], an alternative scenario of the formation of an inward budding MVBs recruiting SMVs at the plasma membrane is unlikely. No such structures or observations were detected in the control plants agroinfiltrated in parallel with only the empty-vector expression cassette plus p22 VSR (Fig 6F). In addition, the plasma membrane appeared to be in close proximity to the cell wall along the entire border of the control plant cells, without any areas of separation such as the vesicle-containing areas in the p33-expressing cells.

Interestingly, the sequences in the p33 CD, including the predicted amphipathic helix, appear to be involved in protein oligomerization and vesicle formation. Previously, we showed that the 30 amino acids at the N-terminal region determine self-interaction of p33 [32]. In this work, we found that a GFP-tagged mutant containing a deletion of these residues, GFP:p33ΔN30, exhibited a drastically different intracellular localization, compared to that of the GFP-tagged wild type p33. GFP:p33ΔN30 was not localized to cellular membranes nor within vesicular bodies but instead formed large irregular inclusion-like structures (Fig 7). Furthermore, our earlier study demonstrated that a mutation of the p33 tyrosine residue 124 to glycine (RFVIRVKAVPASMRG**G**YS) in the predicted cytoplasmic amphipathic helix resulted in the re-localization of the mutated GFP:p33 to amorphous cytoplasmic aggregates, rather than to vesicles or the plasma membrane [22]. The lowered hydrophobicity resulting from the tyrosine to glycine mutation reduces the amphipathicity of the helix. Altogether, these observations suggest that the properties of p33 such as oligomerization and possession of an intact amphipathic helix are needed for the proper localization of the protein on the cellular membranes, and that their alteration will result in a protein that would not be expected to function as a viroporin.

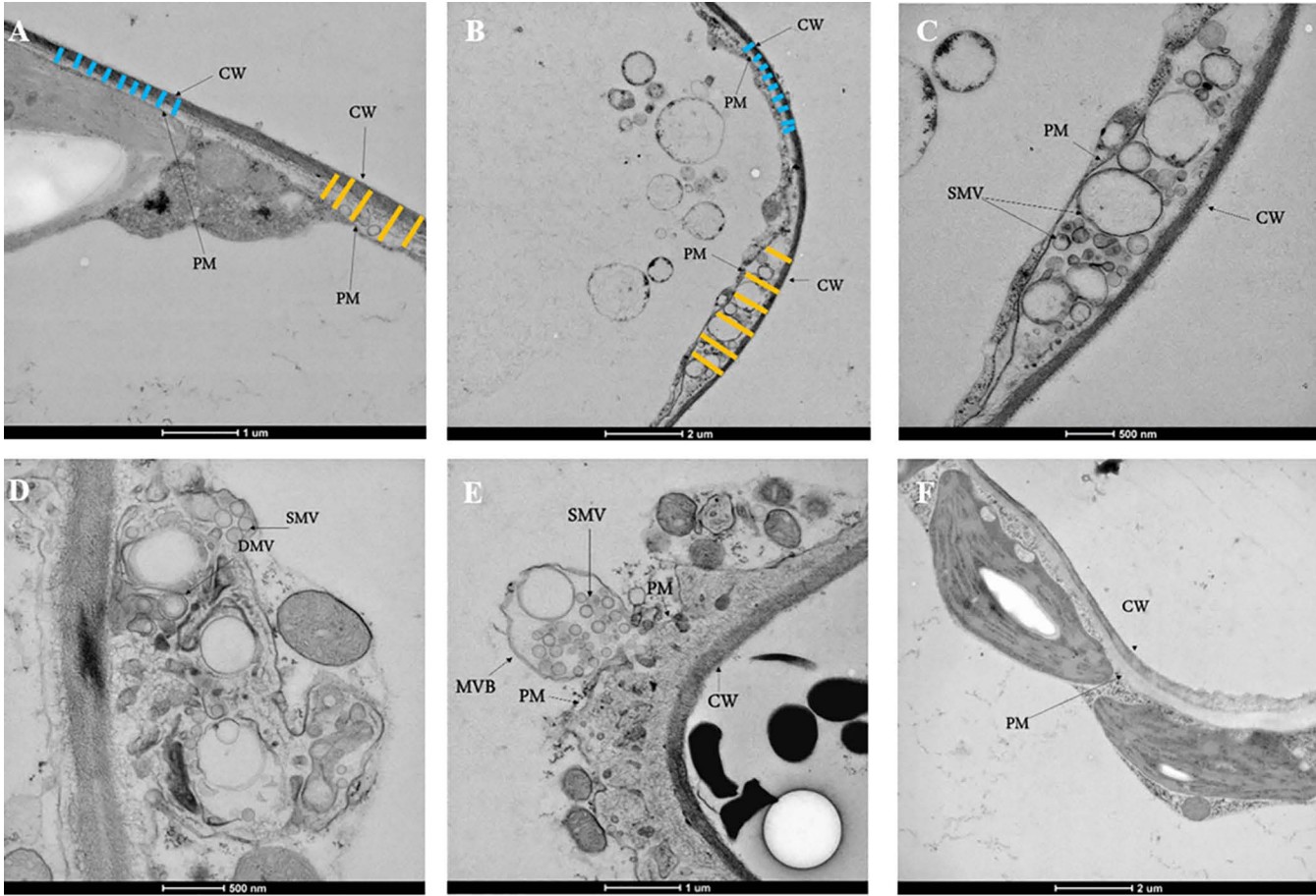

**Fig 6. TEM observations of the vesicular structures and cell wall-plasma membrane interface ultrastructure of *N. benthamiana* epidermal cells in p33-expressing and control cells.** A-B) Wild type p33-expressing cells exhibited areas where the plasma membrane (PM) moved further away from the cell wall (CW), and the space between the PM and the CW became occupied with vesicles (orange bars), compared to other areas of the cell (blue bars). C) A higher magnification of the vesicles in (B), exhibiting what appear to be SMVs ranging between 50-1000 nm in diameter. D) Another example of an extracellular space filled with SMVs and an example of a DMV. E) An MVB filled with SMVs that appears to open into the extracellular space between the PM and CW. F) A control cell expressing only an empty-vector expression cassette and the p22 VSR in which the PM is in close proximity to the CW, and no vesicles or MVBs are observed in the space between the PM and the CW.

## Plasmolysis reveals novel p33 localization

Given that the p33-assocsiated vesicles were found situated between the plasma membrane and the cell wall, the GFP:p33 localization was next observed under the plasmolysis conditions to separate the plasma membrane from the cell wall. GFP:p33 strongly and consistently localized to the Hechtian strands, the linkers between the plasma membrane and the cell wall of plant cells (Fig 8A).

With regards to the GFP:p33-associated vesicles, plasmolysis revealed many peculiar observations. Firstly, GFP:p33-lined vesicles appeared to separate from the plasma membrane in a sort of "budding" motion in a matter of minutes (Fig 8B). It cannot be discerned whether this was a consequence of plasmolysis or an active budding process, which involves p33. We believe, however, that the structures observed with p33 resemble some stages of the budding structures formed by the IAV M2 viroporin [54]. Interestingly, in our earlier study with the GFP:p33-expressing *N. benthamiana* protoplasts [22], GFP-labelled extracellular vesicle-like structures were observed in the association with the surface of the

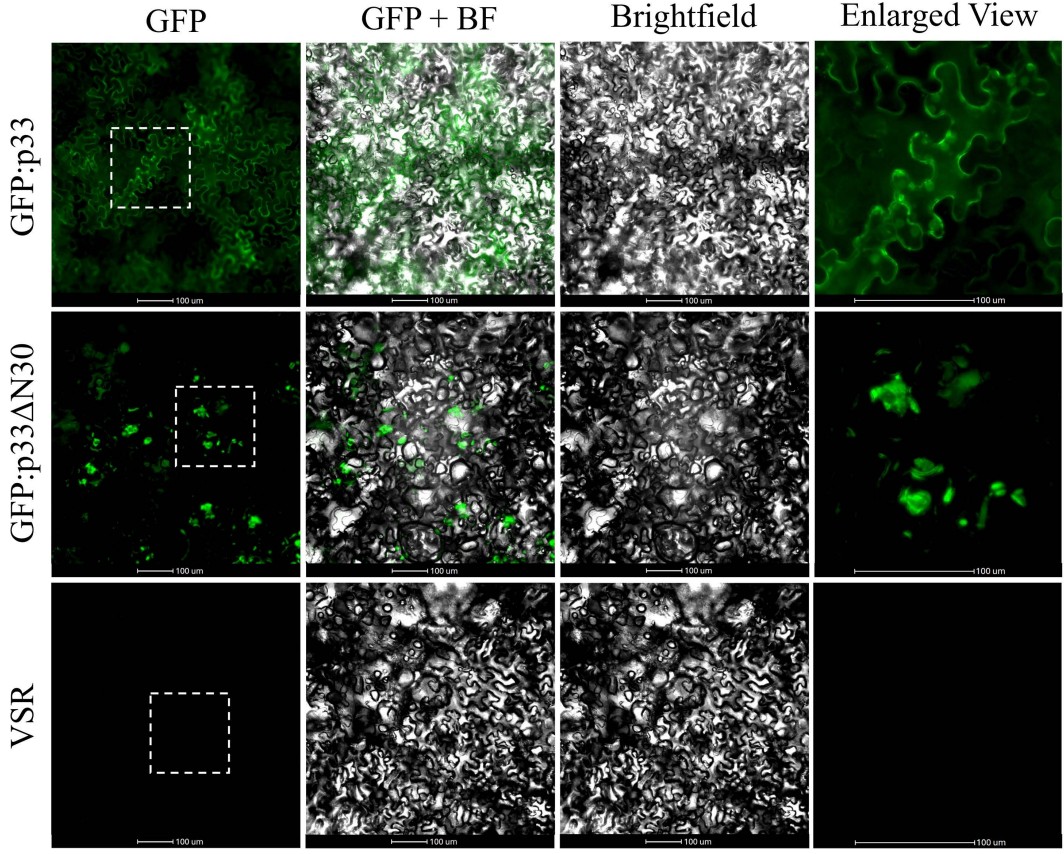

**Fig 7. The p33 deletion mutant GFP:p33ΔN30 is localized in cytoplasmic aggregates compared to the plasma membrane and vesicle local-ization of GFP:p33 when ectopically expressed in *N. benthamiana* epidermal cells.** Epidermal cells were agroinfiltrated with a GFP:p33 + p22, GFP:p33ΔN30 + p22, or just p22 VSR expressing construct and visualized at six dpi using an epifluorescence microscope. The images in the enlarged view column represent a close-up view of the regions within the box in the GFP column. The scale bar represents 100 microns. VSR = viral suppressor of silencing (p22) only. BF = Brightfield.

GFP:p33-expressing protoplasts. Secondly, after a few minutes passed under plasmolysis, punctate GFP:p33 deposits in the cell wall, which were stationary as the cell plasma membrane receded, slowly ballooned out into vesicles (Fig 8C). What appeared as small dots on the cell wall at the beginning of imaging had already grown into small vesicles after 11 minutes. This observation was quite intriguing as the plasma membrane had already receded, so the vesicles did not originate from it.

In summary, p33 was demonstrated to be a viroporin in electrophysiologic assays, induced the accumulation of different vesicular structures in extracellular pockets between the plasma membrane and the cell wall, and localized within budding vesicles that are reminiscent of those induced by Class I viroporins.

## Discussion

Very recently, two proteins encoded in the genome of plant viruses - the potyvirus 6K1 protein [12] and the rhabdovirus P9 protein [13] - have been reported as viroporins. The 6K1 protein was shown to increase cell permeability in *Escherichia coli*, and both 6K1 and P9 could rescue potassium uptake deficiency in yeast. However, these two reports lacked any electrophysiological ion conductance assays typically employed in order to classify a viral protein as a viroporin [14,59].

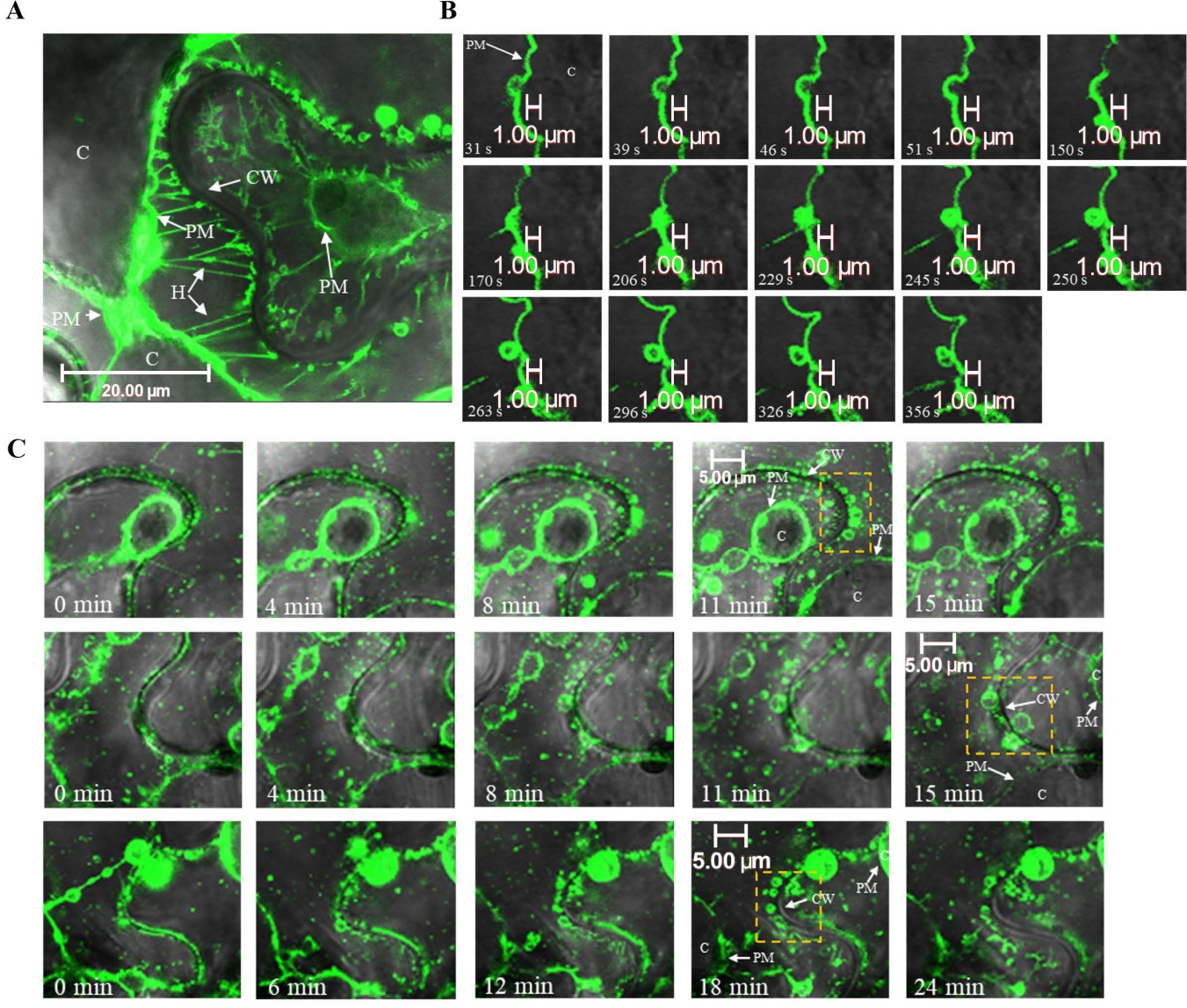

**Fig 8. Plasmolysis reveals novel p33 characteristics in *N. benthamiana* cells.** A) GFP:p33-expressing cells at four dpi under plasmolysis revealed GFP:p33 localization to the Hechtian strands. B) Transiently expressed GFP:p33 localized within a vesicle budding from the plasma membrane under plasmolysis conditions in *N. benthamiana* epidermal cells at four dpi. C) GFP:p33-containing vesicles appeared to grow from punctate GFP:p33 deposits in the cell wall. The dashed boxed regions depict these vesicles. White arrows indicate the plasma membrane (PM), cell wall (CW), and Hechtian strands (H). C: cytoplasm.

Thus, ion conductivity across eukaryotic membranes had not yet been demonstrated electrophysiologically for a plant viral viroporin. In this study, we provide analysis of the p33 structure coupled with the electrophysiological assays to support that this protein is a *bona fide* viroporin encoded by a plant virus. Consequently, this study represents the third report of a viroporin encoded by a plant virus, and the first demonstration of ion conductance across eukaryotic membranes for a plant virus viroporin.

Remarkably, the findings of our previous work investigating the p33 characteristics and functions in the CTV infection cycle have already uncovered many parallels with viroporins encoded by other viruses (Table 1). Multiple animal virus

**Table 1. Functional characteristics shared among the CTV p33 and other known viroporins.**

| Viroporin functional characteristics | Known viroporins | CTV p33 | References |
|---|---|---|---|
| Triggers or regulates PCD | Yes | Yes | [28] |
| Deletion of the protein gene affects virus pathogenicity | Yes | Yes | [21,23,67] |
| Exhibits self-interaction in Y2H, BiFC, and Co-IP | Yes | Yes | [32] |
| Oligomerizes to form channel-like structures | Yes | Yes | This study |
| Exhibits ion conductance in electrophysiological assays | Yes | Yes | This study |
| Remodels intracellular membranes inducing the formation of vesicular bodies | Yes | Yes | This study; [22] |

viroporins have been shown to remodel intracellular membranes [2]. Furthermore, some viroporins such as the HIV-1 Vpu protein and the IAV M2 protein are involved in budding [6,54]. Interestingly, the p33 protein of CTV has been previously observed to induce the formation of vesicles when transiently expressed in the *N. benthamiana* epidermal cells [22]. Moreover, the ectopic expression of p33 in the *N. benthamiana* leaves triggered programmed cell death (PCD) [28]. Remarkably, regulation of PCD or apoptosis through multiple varying mechanisms has also been reported for many viroporins (reviewed in [4]). Some viroporins that have been reported to induce apoptosis include the 2B proteins produced by viruses in the *Picornaviridae* family [60,61], E, 3a, and 8a proteins encoded by the members of the *Coronaviridae* family [62–65], and the IAV M2 protein [66]. On the other hand, some viroporins have been demonstrated to inhibit apoptosis (reviewed in [4]). Finally, the deletion, mutation, or blocking of viroporins have been widely reported to reduce virus pathogenicity [2,3]. In the case of the CTV p33, the deletion of the corresponding gene from the virus genome resulted in its inability to infect certain hosts and in a significant delay of the virus systemic spread in others [23,24]. Importantly, the CTV ability to infect an extended host range is mediated by the p33 TMD [21].

In this study, we first described the structural and sequence similarities p33 shares with other viroporins such as (1) a lack of sequence similarity to any known protein [14]; (2) the formation of tetrameric homo-oligomers, in accordance with all known viroporins, which oligomerize as tetramers, pentamers, or hexamers [68]; (3) the resemblance of the predicted p33 channel-like structures and pores to the experimentally determined structures of known viroporins [2]; and (4) the presence of the cytoplasmic amphipathic helix and phosphorylated residues within the p33 cytoplasmic tail, which resembles Class I viroporins [4,44,45].

Secondly, we performed TEVC assays in *X. laevis* oocytes, providing electrophysiological evidence for p33's ion channel activity. The p33-injected oocytes showed markedly increased inward currents at decreasing membrane potentials for the two monovalent cations tested, $Na^+$ and $K^+$. Viroporins vary in their selectivity for ions. There are examples of high ion selectivity such as for the IAV M2 protein, which is highly selective for protons [1,69]. Earlier, many other viroporins were suggested to have a selectivity for either monovalent ($Na^+$ or $K^+$) or divalent ($Ca^{2+}$) cations. However, while there remains a selectivity for cations over anions, recent findings suggest that most viroporins exhibit only a weak preference for a particular cation [68]. For example, the hepatitis C virus (species: *Hepacivirus hominis*) p7 protein, the alphavirus 6K proteins, the rotavirus NSP4 proteins, and SARS-CoV E protein all have been shown to conduct both monovalent and divalent cations [70–74]. Finally, using confocal microscopy and TEM, we described previously unknown forms of membrane remodeling induced by p33. We demonstrated the induction of p33-associated MVBs as well as different types of large vesicles in pockets formed between the plasma membrane and the cell wall. The disappearance of these GFP:p33-localizing vesicles when the amphipathic helix is mutated suggests some involvement of this helix in the induction of these vesicles [22]. While membrane remodeling is not unique to viroporins, the induction of vesicles and MVBs have been described for several viroporins [2,56,75]. Closteroviruses have been known to induce DMVs and MVBs, likely derived from the ER, which serve as sites of virus replication during infection [76,77]. The expression of the proteins encoded in ORFs 1a/1b in the genome of a related

closterovirus, lettuce infectious yellows virus (species: *Crinivirus lactucaflavi*), triggered the formation of such vesicles independent of other proteins [78]. More recently, a hundred amino acid-long conserved region "Zemlya" in the polyprotein encoded in ORF 1a that contains a conserved amphipathic helix and induces the formation of smaller ER-derived vesicles was identified [79]. The authors suggested that this region might be involved in biogenesis of replication-associated DMVs and MVBs. With that, the pattern of those vesicles is very dissimilar to the p33-derived vesicles observed here. Given that p33 improves the efficiency of the systemic movement of CTV [23], one hypothesis is that the p33-induced membrane remodeling might play a role in enhancing virus movement. Furthermore, DMVs and MVBs are also formed by the picornavirus and coronavirus viroporins (reviewed in [80]). While these structures were initially suggested to originate from autophagosomes, further research disproved their association with autophagy and, instead, showed an ER- or the Golgi apparatus-related origin [81,82]. This agrees with the reports of p33 localizing to the ER- and Golgi-derived vesicles [22]. While some studies suggest that viral extracellular vesicles could move through the cell wall [83], this possibility is not yet well-understood or widely accepted. Hence, the accumulation of the p33-associated vesicles between the cell wall and the plasma membrane and the observed budding-like phenotype is quite puzzling, in terms of the role they could play in the virus infection cycle. The perplexity of those observation is elevated further by the emerging research suggesting that the Hechtian structure could be involved in cellular sensing and signaling and accumulates various regulatory molecules, including ion channels (reviewed in [84]).

The discovery of the viroporin activity of p33, along with the reports on the potyvirus 6K1 and the rhabdovirus P9 proteins, signifies the beginning of a promising, yet unexplored, avenue in the interactions of plant viruses with their hosts. This novel functional characteristic of plant viral proteins can help explain the mechanisms behind the complexity of plant virus transmembrane proteins. Interestingly, it was suggested that another closterovirus protein could be also a viroporin - a small hydrophobic protein p6 shown to be involved in the virus movement [30]. It is not uncommon for viruses to encode more than one viroporin as reported for coronaviruses [85]. Therefore, it is possible that both the p6 and p33 proteins of CTV could function as viroporins.

The p33 protein is not conserved in all viruses belonging to the family *Closteroviridae* or even among members of the genus *Closterovirus*. For example, two well-studied closteroviruses, beet yellows virus (species: *Closterovirus flavibetae*) and grapevine leafroll-associated virus 2 (species: *Closterovirus vitis*), lack a p33-like ORF positioned between the conserved ORF1b and the p6 coding region. On the other hand, other closteroviruses such as strawberry chlorotic fleck-associated virus (species: *Closterovirus fragariae*) and beet yellow stunt virus (species: *Closterovirus nanobetae*) encode for the p33-like p28 and p30 proteins, respectively [86,87]. Therefore, it is possible that p28 and p30 also function as plant viral viroporins. The conservation of the p33-like viroporins in some members of the *Closterovirus* genus, but not others, is truly intriguing and merits further investigation.

The identification of a novel viroporin encoded by a plant virus enriches the pool of information available for future viroporin research. Analysis of the commonalities and dissimilarities between the viroporins from the growing number of viruses infecting organisms from different kingdoms has the potential to expedite viroporin discovery and classification. Furthermore, viroporins are relatively small in size and minimalist in architecture, compared to many eukaryotic ion channels, which are often large and complex, which makes them useful and practical models to study certain structural aspects shared by eukaryotic ion channels and viroporins [29,88]. Consequently, plant virus viroporins represent a unique model system for studying plant ion channels. Finally, given the importance of viroporins for the virus pathogenicity and the clinical importance of several viroporin-encoding human viruses, viroporins are the focus of many research efforts investigating them as targets for inhibitory drug development and antiviral therapy, with many of the potential drugs being ion channel blockers [4,29,88,89]. The discovery of effective ion channel blocking antiviral drugs, coupled with the possibility of uncovering the yet unexplored role of viroporins in plant virus infections, has the potential to greatly impact plant virology and provide the agricultural industry with a much-needed means to reduce the extensive economic damage these viruses cause.

PLOS Pathogens

## Materials and methods

### Ethics statement

All experimental procedures involving frogs were approved by the University of Florida Institutional Animal Care and Use Committee (the approval number 202002669).

### Generation of constructs

The p22 protein of tomato chlorosis virus was shown to provide strong activity as an VSR and support marked CTV expression from a pCAMBIA binary vector in *N. benthamiana* when the p22 gene was cloned into the same vector as the CTV genome (pCTV-p22; [58]). We, therefore, cloned the p22 gene into a pCass4N-based binary vector, in order to use this construct in co-expression with the p33 protein and enhance the level of the p33 production upon *Agrobacterium*-mediated expression in *N. benthamiana* plants.

The p22 construct was cloned by amplifying a section of a binary vector pCTV-p22 carrying the CTV genome and p22 [58] that excludes the CTV genomic sequence but retains the vector components and the p22 ORF by using the primers p22F (5'-CGGGATTTAAATACTAGTGCGGCC-3') and p22R (5'-CCGGGCCCAACATGGTGG-3'). This PCR product was phosphorylated, self-ligated, and the template vector was degraded in a single step using the KLD enzyme mix (New England Biolabs, Ipswich, MA, United States). The ligation product was transformed first into JM109 *E. coli* competent cells (Promega, Madison, WI, United States) and later into the *Agrobacterium tumefaciens* strain GV3101.

For electrophysiological assays, the complete nucleotide sequence of the p33 gene was amplified using the primers P33-pSGEM-F (5'-CGGGATCCATGTTTGCCTTCGCGAG-3') and P33-pSGEM-R (5'-CCGCTCGAGTCATATAAATAT AATGGCTAATAAACCGC-3'). The sequences were inserted between the *BamH*I and *Xho*I restriction sites of the *X. laevis* expression plasmid pSGEM (obtained from Dr. Michael Hollmann (Ruhr University, Bochum, Germany)) under the T7 promoter.

### Laser-scanning confocal microscopy

The *A. tumefaciens* strain GV3101 cultures transformed with the binary vectors expressing a GFP-tagged p33 (GFP:p33) [21] and p22 were co-infiltrated into six-week-old *N. benthamiana* plants at an $OD_{600}$ of 0.9 and 0.1, respectively. The epidermal cells of the leaves were visualized using Leica TCS SP5 confocal laser scanning microscope system (Leica Microsystems, Wetzlar, Germany) at either four or seven dpi. Plasmolysis was induced by infiltration of leaves with 30% glycerol. Leaf discs were subsequently excised and placed in 30% glycerol on a glass slide before imaging. The GFP signal was excited at 488 nm and the emission was collected at 500–530 nm. Images were captured using a 100X oil objective lens and processed using LAS-X (Leica Microsystems, Wetzlar, Germany) software.

### Transmission electron microscopy (TEM)

Samples of *N. benthamiana* leaf tissue agroinfiltrated with either a p33-expressing or an empty vector were collected at six dpi and processed using high-pressure freeze and freeze substitution (HPF/FS) cryofixation. Four-mm leaf punches were cut into quarters and immersed into 4% paraformaldehyde in phosphate-buffered saline solution, pH 7.2. After buffer washes, the leaf quarters were placed in type A aluminum specimen carriers along with 1μl 1-hexadecene and high-pressure frozen (HPM100 Leica Microsystems, Wetzlar, Germany). The specimen carriers containing HPF frozen leaves were transferred into vials containing $LN_2$ frozen cocktail, 2% $OsO_4$, 0.05% uranyl acetate in anhydrous acetone. The vials containing leaves were then placed into a -192 ºC precooled freeze substitution unit (AFS2 Leica Microsystems, Wetzlar, Germany) and raised to -90 ºC. After 72 hours, the temperature was slowly raised to -20 ºC over a 24-hour period. The samples were then washed three times in cold anhydrous acetone over a 3-hour period. Specimen carriers were removed from the vials and the temperature was raised to 4 ºC over 5 hours. The leaf tissue samples were brought

to room temperature with additional acetone washes followed by Araldite/Embed epoxy resin infiltration containing Z6040 embedding primer (Electron Microscopy Sciences, Hatfield, PA, United States). Ultrathin sections cut to 120 nm were collected onto carbon-coated Formvar 2x1 mm copper slot grid (EMS, Hatfield, PA, United States) and post-stained with 2% aqueous uranyl acetate and lead citrate. Sections were examined with a FEI Tecnai G2 Spirit Twin TEM (FEI Corp., Hillsboro, OR, United States), and digital images were acquired with a Gatan UltraScan 2k x 2k camera and Digital Micrograph software (Gatan Inc., Pleasanton, CA, United States). The TEM work was carried out at the University of Florida Interdisciplinary Center for Biotechnology Research (UF-ICBR) Electron Microscopy core, RRID:SCR_019146.

## Epifluorescence microscopy

*A. tumefaciens* cultures transformed with binary vectors expressing GFP:p33 or GFP:p33ΔN30 [32] along with the p22-expressing cassette, were agroinfiltrated into leaves of six-week-old *N. benthamiana* following the same protocol as described above. *N. benthamiana* leaves infiltrated with *A. tumefaciens* expressing only p22 were used for the negative control. At six dpi, agroinfiltrated leaf tissue samples were excised and mounted on glass slides in water. Imaging was performed using an ECHO Revolve epifluorescence microscope (Discover Echo, San Diego, CA, United States). Images were captured using a 10x objective lens and GFP fluorescence was visualized with laser power set to 50% and exposure time set to 30 milliseconds.

## p33 immunoprecipitation and proteomics

The *A. tumefaciens* strain GV3101 cultures transformed with the binary vectors expressing p33 [19] and p22 were co-infiltrated into six-week-old *N. benthamiana* plants at $OD_{600}$ of 0.9 and 0.1, respectively. At six dpi, 1 g of p33-expressing leaves was ground in liquid nitrogen into a fine powder. The powder was added to the extraction buffer (50 mM Tris-HCl pH 7.5, 1 mM EDTA, 150 mM NaCl, 5% glycerol, 0.5% Triton X-100, 1X Halt Protease and Phosphatase Inhibitor Cocktail (Thermo Fisher Scientific)), vortexed, and incubated on a rotating shaker at 4 °C for 60 minutes. After centrifugation at 3000 x g for 10 min at 4 °C, the supernatant was transferred into a new tube. Dynabeads Protein A Immunoprecipitation Kit (Thermo Fisher Scientific) was used to link magnetic protein A beads to an anti-p33 antibody [21], which was used to immunoprecipitate p33 from the total lysate supernatant following the manufacturer's instructions. After immunoprecipitation and washing, the beads were incubated in SDS loading buffer (50 mM Tris-HCl pH 6.8, 2% SDS, 2 mM EDTA, 5% glycerol, 0.5 ppm bromophenol blue, and freshly added 100 mM DTT) supplemented with 8M urea at 70 °C for 10 min. An aliquot of the eluted fractions was analyzed on an SDS-PAGE gel by staining with Bio-Safe Coomassie Stain (Bio-Rad, Hercules, CA, United States) and western blotting. In the crude extraction method, two microcentrifuge tube cap-sized leaf discs were directly macerated in the 8M urea SDS loading buffer and loaded on the gel.

For proteomic analysis, after confirming the presence of strong bands corresponding to the p33's molecular weight, 50 μL of the eluted protein was loaded on a 12% Mini-PROTEAN TGX Precast Gel (Bio-Rad, Hercules, CA, United States) and electrophoresed for 1 cm. The gel was stained using Bio-Safe Coomassie Stain (Bio-Rad, Hercules, CA, United States), and the gel pieces containing the stained p33 bands were excised and submitted to the UF-ICBR Proteomics & Mass Spectrometry Core Facility, RRID:SCR_019151, for trypsin/Lys-C digestion, phosphopeptide enrichment, and the identification of phosphorylated peptides through Orbitrap-Fusion liquid chromatography with tandem mass spectrometry (LC-MS/MS).

## Western blotting

The same western blotting method was utilized for the p33 protein samples purified from *N. benthamniana* or *X. laevis*. The samples from *N. benthamiana* were prepared as described above. For *X. laevis*, each oocyte injected with the RNA transcript complementary to the p33-gene (cRNA) was collected in a separate 1.5 mL tube and lysed by pipetting in SDS

loading buffer (50 mM Tris-HCl pH 6.8, 2% SDS, 2 mM EDTA, 5% glycerol, 0.5 ppm bromophenol blue) supplemented with 8M urea. All samples were run on 12% Mini-PROTEAN TGX Precast Gel (Bio-Rad, Hercules, CA, United States) and electro-transferred to polyvinylidene fluoride membranes. Western blot was carried out with the anti-p33 rabbit polyclonal antibody [21] and 1:20000 dilution of the goat anti-rabbit IgG H L (HRP) secondary antibody (Abcam) after blocking with 5% non-fat milk in tris buffered saline with Tween-20. The blots were treated with Clarity Western ECL Substrate (Bio-Rad, Hercules, CA, United States) and visualized on ChemiDoc XRS+ system with Image Lab software (Bio-Rad, Hercules, CA, United States).

### Electrophysiological assays

The coding sequence of the p33 gene flanked by the *Xenopus* β-Globin gene 5'- and 3'- untranslated regions was cloned into the pSGEM expression vector kindly provided by Dr. Roger Papke (Department of Pharmacology and Therapeutics, University of Florida), downstream of a T7 promoter. The vectors were linearized by *Pac*I and purified using QIAquick PCR Purification Kit (Qiagen, Germantown, MD, United States). The capped complement RNA (cRNA) was transcribed using the mMESSAGE mMACHINE T7 transcription kit (Ambion, Austin, TX, United States) and purified using the Monarch RNA Cleanup Kit (New England Biolabs, Ipswich, MA, United States).

The oocytes were prepared as described in [90]. The frogs were maintained in the University of Florida Animal Care Service facility, and all procedures were approved by the University of Florida Institutional Animal Care and Use Committee (approval number 202002669). Briefly, the frogs were first anesthetized for 15–20 min in 1.5 L frog tank water supplemented with 1 g of MS-222 buffered with sodium bicarbonate. Oocytes were surgically collected from mature female *X. laevis* frogs (Nasco, Ft. Atkinson, WI, United States) and subjected to 1.4 mg/mL type 1 collagenase (Worthington Biochemicals, Freehold, NJ, United States) treatment for 2–4 hours at room temperature in calcium-free Barth's solution (88 mM NaCl, 1 mM KCl, 2.38 mM NaHCO$_3$, 0.82 mM MgSO$_4$, 15 mM HEPES, and 12 mg/L tetracycline, pH 7.6) for the removal of the ovarian tissue and the follicular layers. Approximately 27 ng of cRNA in 50 nL water was injected into each stage V *X. laevis* oocyte. Oocytes were maintained in Barth's solution supplemented with 0.32 mM Ca(NO$_3$)$_2$ and 0.41 mM CaCl$_2$, and recordings were carried out 8 days after injection. TEVC experiments were conducted using OpusXpress 6000A (Molecular Devices, San Jose, CA, United States) [91]. Both the voltage and current electrodes were filled with 3 M KCl and had a resistance of 0.5–3 mega ohms. Oocytes were voltage-clamped at room temperature at -40 mV for Na$^+$ assays and -20 mV for K$^+$ assays. The oocytes were perfused with Ringer's solution (115 mM NaCl, 2.5 mM KCl, 1.8 mM CaCl$_2$, and 10 mM HEPES, 1 μM atropine, pH 7.2) for Na$^+$ assays or 50 mM K$^+$ Ringer's (67.5 mM NaCl, 50 mM KCl, 1.8 mM CaCl2, 10 mM HEPES, 1 μM atropine, pH 7.2) for K$^+$ assays at 2 mL/min. For K$^+$ assays, the clamped oocytes were perfused with High K$^+$ ringers for 5 min prior to beginning the experiments.

The standard voltage-clamp protocol consisted of 2s-long rectangular voltage steps, with 2s-long intervals between them, from +50 to -150 mV in 10 mV increments applied from a holding voltage of -40 mV for Na$^+$ assays and -20 mV for K$^+$ assays. For statistical analysis at each voltage step, baseline readings were taken between steps at the end of the 2 s interval, and the current amplitude was measured between 1.5-2 s into the voltage steps. Data were collected at 500 Hz, filtered at 100 Hz, and analyzed by Clampfit (Molecular Devices, San Jose, CA, United States) and Excel (Microsoft, Redmond, WA, United States). Data were expressed as the means of ± SEM from at least seven oocytes for each experiment and plotted with Excel (Microsoft, Redmond, WA, United States). Statistical analyses were performed by using Student's t test.

### Supporting information

**S1 Fig. Western blot of purified p33 protein transiently expressed in *N. benthamiana* displaying protein bands which correspond to the expected sizes of p33 monomers, dimers, and tetramers.**
(TIF)

## Acknowledgments

We would like to express our gratitude to Drs. Elias Bassil, Sixue Chen, and Bin Liu for their valuable advice regarding certain aspects of this work. We would also like to thank Karen Kelly at the University of Florida ICBR-EM core for technical assistance with the electron microscope.

## Author contributions

**Conceptualization:** Vicken Aknadibossian, Svetlana Y. Folimonova.

**Data curation:** Vicken Aknadibossian, Clare Stokes, Roger L. Papke, Hao Wei Teh, Svetlana Y. Folimonova.

**Formal analysis:** Vicken Aknadibossian, Clare Stokes, Roger L. Papke, Hao Wei Teh, Ying Wang, Svetlana Y. Folimonova.

**Funding acquisition:** Svetlana Y. Folimonova.

**Investigation:** Vicken Aknadibossian, Clare Stokes, Roger L. Papke, Hao Wei Teh, Ying Wang, Svetlana Y. Folimonova.

**Methodology:** Vicken Aknadibossian, Clare Stokes, Roger L. Papke, Svetlana Y. Folimonova.

**Project administration:** Svetlana Y. Folimonova.

**Resources:** Svetlana Y. Folimonova.

**Supervision:** Svetlana Y. Folimonova.

**Validation:** Vicken Aknadibossian, Roger L. Papke, Hao Wei Teh, Ying Wang, Svetlana Y. Folimonova.

**Visualization:** Vicken Aknadibossian, Clare Stokes, Hao Wei Teh, Svetlana Y. Folimonova.

**Writing – original draft:** Vicken Aknadibossian, Svetlana Y. Folimonova.

**Writing – review & editing:** Vicken Aknadibossian, Clare Stokes, Roger L. Papke, Hao Wei Teh, Ying Wang, Svetlana Y. Folimonova.

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
