## [Decision Letter · Decision Letter 0]

29 Jan 2025

Plant virus viroporins on the rise: citrus tristeza virus protein p33 is a novel viroporin encoded by a plant virus

PLOS Pathogens

Dear Dr. Folimonova,

Thank you for submitting your manuscript to PLOS Pathogens. After careful consideration, we feel that it has merit but does not fully meet PLOS Pathogens's publication criteria as it currently stands. Therefore, we invite you to submit a revised version of the manuscript that addresses the points raised during the review process.

Please submit your revised manuscript within 60 days Mar 30 2025 11:59PM. If you will need more time than this to complete your revisions, please reply to this message or contact the journal office at plospathogens@plos.org. Please include the following items when submitting your revised manuscript:

We look forward to receiving your revised manuscript.

Kind regards,

Peter D. Nagy

Academic Editor

PLOS Pathogens

Shou-Wei Ding

Section Editor

PLOS Pathogens

Editor-in-Chief

PLOS Pathogens

orcid.org/0000-0003-2946-9497

Michael Malim

Editor-in-Chief

PLOS Pathogens

orcid.org/0000-0002-7699-2064

**Additional Editor Comments:**

The reviewers appreciated the identification of a novel plant virus viroporin and its effect on membrane potential and novel membranous structures formation, including extracellular vesicles. The reviewers made several valuable recommendations, including on analyzing p33 mutants that would significantly strengthen the paper.

**Journal Requirements:**

- ® on pages: 28 line 619, and 29 lines 632 and 646

- TM on pages: 28 lines 607, 610, 616, 619, and 620, 29 lines 632, 636, 637, 638, and 642.

4) We note that your Data Availability Statement is currently as follows: "All data are provided in the manuscript.". Please confirm at this time whether or not your submission contains all raw data required to replicate the results of your study. Authors must share the “minimal data set” for their submission. PLOS defines the minimal data set to consist of the data required to replicate all study findings reported in the article, as well as related metadata and methods (https://journals.plos.org/plosone/s/data-availability#loc-minimal-data-set-definition).

1)  If the funders had no role in your study, please state: "The funders had no role in study design, data collection and analysis, decision to publish, or preparation of the manuscript."

6) Your current Financial Disclosure states, "This work was supported by the National Science Foundation (Grant Numbers MCB-1615723 and MCB-2316587 to S. Y. Folimonova) and the United States Department of Agriculture (USDA) National Institute of Food and Agriculture (NIFA), Hatch Project FLA-PLP-006024. "

However, your funding information on the submission form indicates only this funder " National Science Foundation". Please ensure that the funders and grant numbers match between the Financial Disclosure field and the Funding Information tab in your submission form. Note that the funders must be provided in the same order in both places as well.

Please indicate by return email the full and correct funding information for your study and confirm the order in which funding contributions should appear. Please be sure to indicate whether the funders played any role in the study design, data collection and analysis, decision to publish, or preparation of the manuscript.

**Reviewers' Comments:**

Reviewer's Responses to Questions

**Part I - Summary**

Reviewer #1: This manuscript shows that Citrus tristeza virus protein 33 (P33) is a viroporin (also called viral ion channel forming proteins, or viral channel forming proteins). Results are in formative as they establish the foundation to look for and understand the role of viroporins in plant viruses. However, the manuscript needs to be restructured to highlight the novelty without taking credit away from previous publications that arrived to the same conclusion through different methods.

Reviewer #2: The manuscript by Aknadibossian et al. describes a highly attractive viroporin encoded by a plant virus. The authors provide evidence suggesting that the p33 protein shares many similarities with known class I viroporins. Bioinformatics tools predict several features of p33 that are consistent with known viroporins.

Reviewer #3: The manuscript “Plant virus viroporins on the rise: citrus tristeza virus protein p33 is a novel viroporin encoded by a plant virus” by Aknadibossian et al reported that p33 of CTV functions a viroporin based on the structural and function similarities to those well-studied viral proteins of human viruses: p33 formed dimer and tetramer when expressed in N. benthamiana cells; showed ion channel activity in the two-electrode voltage-clamp assay; caused extensive membrane rearrangement including vesicle in the cytoplasm, PM, and between the PM and Cell wall. While two papers have recently been published in Plant Cell and PNAS on two plant viral proteins as viroporin, p33 was demonstrated to have ion channel activity by using the two-electrode voltage-clamp assay. The manuscript was well-written and its content merits to publish in PLoS Pathogens with further verification.

**Part II – Major Issues: Key Experiments Required for Acceptance**

Reviewer #1: There are two previous reports about plant virus viroporins. Thus, the statement on lines 31-32 , and equivalent everywhere in the manuscript, needs to be revised and remove the idea that this is the first report. This is despite the argument in lines 91-93. The conclusion does not change, this is not the first report of a viroporin in a plant virus.

Reviewer #2: Some experimental details still need further validation. For instance, the reliability of the oligomeric state bands observed in SDS-PAGE under non-denaturing conditions, the composition of the solutions and blockers used in electrophysiological experiments, and the identification of vesicle-like structures and Hechtian strands observed under confocal microscopy. Also�it would be interesting to know how the vesicle structures formed by the p33 protein are correlated with its ion channel activity.

Reviewer #3: The conclusions that p33 is a viroporin will be substantially strengthened if mutations that affect the pore structure or ion channel activity can be confirmed. The effects of such mutations on oligomerization, ion channel, among other assays, should be tested. The same is true for the projected amphipathic helix. Key amino acids can be targeted to disrupt helix or amphipathicity to confirm such structure plays a key role in the formation of pore, oligomerization, ion channel, etc.

Fig. 4 A known viroporin should be included as a positive for comparison purposes: is p33 has strong or minimal activity compared to other well-characterized viroporins.

**Part III – Minor Issues: Editorial and Data Presentation Modifications**

Reviewer #1: 1. The title needs to be re-structured, CTV is a plant virus.

2. Current binomial names for virus species should be used.

3. Figure legends are dispersed within the text: Lanes 232-237 are misplaced. Same for lanes 270-279, Fig 4 legend, and others.

4. Tatineni et al., 2012 is not found in references.

5. Figure 5 legend needs to be revised for clarity.

Reviewer #2: Page 8, Line 157-158, Figure 1A, how to identify the cytoplasmic domain (CD) and external domain (ED) of the known viroporins and p33?

Page 8, Figure 1A, please specify which tools were used to predict the TMD.

Page 8, Line 183-185, Figure 1B, I noticed that the protein extraction process did not undergo sufficient denaturation (e.g., treatment with β-ME). If the protein structure is not fully unfolded, can its size be used as a reference in SDS-PAGE electrophoresis? Please provide native electrophoresis results or other evidence to support the conclusion that p33 is in a oligomeric state.

Page 8, Figure 1C, please add a loading control to this figure. Besides, please specify the meaning of the numbers at the top of the figure in the figure legend.

Page 8, Figure 1, what is the difference in the protein extraction methods between Figure 1A and 1B? Please provide the details in the Methods section.

Page 15, Line 315, LaCl3 is merely Ca2+ channel inhibitor, how can the influence of calcium-activated chloride channel be eliminated?

Page 15, Figure 4, according to other studies, the control group injected with water should show no current changes, why does the control group in your results also exhibit current changes

Page 15, Line 330, when conducting Na+ or K+ assays, retaining the presence of the other ion in the solution, even at a lower concentration, is not rigorous. Additionally, the influence of other ions, such as Ca2+ and Cl-, has not been excluded.

Page 15, Line 331, please provide the reference about “Most viroporins that conduct potassium also conduct sodium”, thank you.

Page 16, Line 357-359, “It was postulated...to reach the” is an incomplete sentence.

Page 17, Line 363, please write out the abbreviation on first use.

Page 17, Line 369, how to determine the vesicle structure? Are there corresponding markers? At the same time, a negative control is needed to exclude the role of p22 in vesicle formation.

Page 18, Figure 6, why not use immunogold labeling to confirm the presence of p33 on the vesicles? Is the formation of vesicles dependent solely on the viral p33 protein?

Page 19, Line 412-413, Figure 7, how to determine the Hechtian strands? Are there corresponding markers?

Page 29, Line639, the blocker LaCl3 mentioned earlier were not described in the methods section, please provide additional details.

Reviewer #3: The title starts with “Plant virus viroporins on the rise”, which gives a false impression that this is a review paper.

L359, missing a word after “the”.

PLOS authors have the option to publish the peer review history of their article (what does this mean? ). If published, this will include your full peer review and any attached files.

**Do you want your identity to be public for this peer review?** For information about this choice, including consent withdrawal, please see our Privacy Policy .

Reviewer #1: **Yes: ** Hernan Garcia-Ruiz

Reviewer #2: No

Reviewer #3: No

**Figure resubmission:**

**Reproducibility:**



---

## [Decision Letter · Decision Letter 1]

11 May 2025

The citrus tristeza virus p33 protein functions as a viroporin

PLOS Pathogens

Dear Dr. Folimonova,

Thank you for submitting your manuscript to PLOS Pathogens. Initially, we invited two reviewers who reviewed the previous version unfortunately one of them is not available. So, we invited a new reviewer who is an expert on viroporin. As you can see comments below, this reviewer is also very positive with your findings but also suggested key experiments to improve. After careful consideration, we feel that this revision has improved but still does not fully meet PLOS Pathogens' publication criteria as it currently stands. Therefore, we invite you to submit a revised version of the manuscript that addresses the points raised during the review process.

Please submit your revised manuscript within 60 days Jul 10 2025 11:59PM. If you will need more time than this to complete your revisions, please reply to this message or contact the journal office at plospathogens@plos.org. Please include the following items when submitting your revised manuscript:

We look forward to receiving your revised manuscript.

Kind regards,

Aiming Wang, Ph.D

Academic Editor

PLOS Pathogens

Shou-Wei Ding

Section Editor

PLOS Pathogens

Editor-in-Chief

PLOS Pathogens

orcid.org/0000-0003-2946-9497

Editor-in-Chief

PLOS Pathogens

orcid.org/0000-0002-7699-2064

**Reviewers' Comments:**

Reviewer's Responses to Questions

**Part I - Summary**

Reviewer #3: The authors have addressed most of my requests, though the retirement of a coauthor prevents them from performing TEVC assays to include a well-studied viroporin.

I have no further request.

Reviewer #4: The manuscript by Aknadibossian et al. reports that the p33 of citrus tristeza virus (CTV) is a function viroporin. They first compared the structural similarities between p33 and known viroporins, then tested the viroporin activity using the electrode clamp assay, and finally investigated its subcellular localization and membrane remodelling ability. The discovery of p33 as the third plant virus-encoded viroporin is novel and interesting. Although the electrode clamp assay clearly showed the viroporin activity of p33, some experiments of this study are not consistent or are not related to the viroporin activity in my opinion (see below).

**Part II – Major Issues: Key Experiments Required for Acceptance**

Reviewer #3: (No Response)

Reviewer #4: 1, the only experiment directly related to the ion channel activity is the electrode clamp assay. Although reliant, the use of only p33 wide-type but no p33 mutants was used is unacceptable (also noted by the reviewers in the previous review). The authors explained the availability of the equipment for the electrode clamp assay, which is acceptable. I think it is necessary to test the viroporin activity using the yeast complementation assay with the K+ uptake deficient yeast strain CY162 (commercially available).

2, The results in Fig. 3A and B are not related to the putative a-helix forming the channel, which is located at the C-terminus and should therefore be deleted.

3, The relationship between viroporin activity and phosphorylation is too far-fetched unless the authors have shown that p33 mutations with either a D or A substitution of S8 have increasesed or decreased viroporin activity. Therefore, Fig. 3C can also be deleted.

4, Fig. 5 to 7 are also not strongly related to the viroporin activity; however, analysis of the localization of p33 during CTV infection may shed light on the function of p33 viroporin during viral infection.

**Part III – Minor Issues: Editorial and Data Presentation Modifications**

Reviewer #3: (No Response)

Reviewer #4: 1, please note that the arrangement of CD and ED in p33 is opposite to that of IAV M2, HIV Vpu and SAR Cov E in Fig. 1A.

2, lines 179-183, why does the addition of SGS and 8M urea fail to disrupt p33 tetramers? Is it possible that disulfide bonds were involved in the formation of p33 tetramers? This can be confirmed by the addition of DTT.

3, line 207, please delete the context in parentheses as it has already been described in line 189.

4, the scale bars in Fig. 5-7 are of different colors, thicknesses, location, and sizes, so keep consistent.

5, Please check the scale bars between panels A and B in Fig. 5.

PLOS authors have the option to publish the peer review history of their article (what does this mean? ). If published, this will include your full peer review and any attached files.

**Do you want your identity to be public for this peer review?** For information about this choice, including consent withdrawal, please see our Privacy Policy .

Reviewer #3: No

Reviewer #4: **Yes: ** Xiaofei Cheng

**Figure resubmission:**

**Reproducibility:**



---

## [Decision Letter · Decision Letter 2]

15 Nov 2025

Dear Dr. Folimonova,

We are pleased to inform you that your manuscript 'The citrus tristeza virus p33 protein functions as a viroporin' has been provisionally accepted for publication in PLOS Pathogens.

Best regards,

Aiming Wang, Ph.D

Academic Editor

PLOS Pathogens

Shou-Wei Ding

Section Editor

PLOS Pathogens

Sumita Bhaduri-McIntosh

Editor-in-Chief

PLOS Pathogens

orcid.org/0000-0003-2946-9497

Michael Malim

Editor-in-Chief

PLOS Pathogens

orcid.org/0000-0002-7699-2064

Reviewer Comments (if any, and for reference):

Reviewer's Responses to Questions

**Part I - Summary**

Reviewer #4: I believe discover the p33 has a viroporin activity is novel to the field of plant virology and will have a broad interest to all plant virologists.

**Part II – Major Issues: Key Experiments Required for Acceptance**

Reviewer #4: The authors have answered all my concerns and I have no more question this time.

**Part III – Minor Issues: Editorial and Data Presentation Modifications**

Reviewer #4: No.

PLOS authors have the option to publish the peer review history of their article (what does this mean? ). If published, this will include your full peer review and any attached files.

**Do you want your identity to be public for this peer review?** For information about this choice, including consent withdrawal, please see our Privacy Policy .

Reviewer #4: No

---

## [Editor Report · Acceptance letter]

Dear Dr. Folimonova,

We are delighted to inform you that your manuscript, "The citrus tristeza virus p33 protein functions as a viroporin," has been formally accepted for publication in PLOS Pathogens.

Best regards,

Sumita Bhaduri-McIntosh

Editor-in-Chief

PLOS Pathogens

orcid.org/0000-0003-2946-9497

Michael Malim

Editor-in-Chief

PLOS Pathogens

orcid.org/0000-0002-7699-2064